# Re-examining the role of *Drosophila* Sas-4 in centrosome assembly using two-colour-3D-SIM FRAP

Paul T Conduit[1,2]*[†], Alan Wainman[2†], Zsofia A Novak[2†], Timothy T Weil[1], Jordan W Raff[2]*

[1]Department of Zoology, University of Cambridge, Cambridge, United Kingdom;
[2]Sir William Dunn School of Pathology, University of Oxford, Oxford, United Kingdom

**Abstract** Centrosomes have many important functions and comprise a 'mother' and 'daughter' centriole surrounded by pericentriolar material (PCM). The mother centriole recruits and organises the PCM and templates the formation of the daughter centriole. It has been reported that several important *Drosophila* PCM-organising proteins are recruited to centrioles from the cytosol as part of large cytoplasmic 'S-CAP' complexes that contain the centriole protein Sas-4. In a previous paper (*Conduit et al., 2014b*) we showed that one of these proteins, Cnn, and another key PCM-organising protein, Spd-2, are recruited around the mother centriole before spreading outwards to form a scaffold that supports mitotic PCM assembly; the recruitment of Cnn and Spd-2 is dependent on another S-CAP protein, Asl. We show here, however, that Cnn, Spd-2 and Asl are not recruited to the mother centriole as part of a complex with Sas-4. Thus, PCM recruitment in fly embryos does not appear to require cytosolic S-CAP complexes.

*For correspondence:
ptc29@cam.ac.uk (PTC);
jordan.raff@path.ox.ac.uk (JWR)

[†]These authors contributed equally to this work

**Competing interests:** The authors declare that no competing interests exist.

## Results

### The centrosomal recruitment of Sas-4, Cnn and Spd-2 differs in space and time

Centrosomes are crucial cell organisers (*Nigg and Raff, 2009*; *Arquint et al., 2014*; *Chavali et al., 2014*; *Reina and Gonzalez, 2014*; *Stinchcombe and Griffiths, 2014*). We previously showed that Cnn and Spd-2 are initially recruited around mother centrioles and then spread outward to form an extended pericentriolar material (PCM) scaffold (*Conduit et al., 2010*, *2014a*, *2014b*) (Note that we define 'recruitment' here as when a new protein molecule is added into the centrosome from the cytosol, irrespective of whether this molecule replaces an existing molecule or adds to the existing pool of molecules). Cnn has previously been identified as part of a multi-protein 'S-CAP' complex, which pre-assembles in the cytosol with the centriole protein Sas-4 before being recruited into the centrosome via a Sas-4–centriole interaction (*Gopalakrishnan et al., 2011*, *2012*; *Zheng et al., 2014*). We reasoned, therefore, that Cnn and Spd-2 molecules might initially be recruited to the centrioles in S-CAP complexes, but could then be released from the centriolar-Sas-4 to spread outwards through the PCM. To test this possibility we compared the spatiotemporal centrosomal recruitment of Sas-4 to Cnn or Spd-2 in living *Drosophila* syncytial embryos, where S-CAP complexes were initially identified (*Gopalakrishnan et al., 2011*). These embryos cycle rapidly between S- and M-phases with no gap phases, and the mother centrioles organise large amounts of PCM throughout both S- and M-phases; during S-phase, each mother centriole also assembles a new daughter centriole (*Figure 1A*).

We co-expressed Sas-4-mCherry with either GFP-Cnn or Spd-2-GFP and performed two-colour Fluorescence Recovery After Photobleaching (two-colour FRAP) on a spinning disk confocal

**Figure 1**. The centrosomal recruitment of Sas-4, Cnn and Spd-2 differ in space and time. (**A**) A schematic illustration of the centrosomal events that occur during S-phase in *Drosophila* syncytial embryos. The mother centriole ('m') constantly organises pericentriolar material (PCM, green) and also templates the formation of a new daughter centriole ('d') that grows throughout S-phase. (**B**) Images show how GFP-Cnn (top row, *green* in bottom row) and Sas-4-mCherry (middle row, *red* in bottom row) fluorescence signals recover after photobleaching. Time in seconds before and after photobleaching (t = 0 s) is shown in the top right of each panel. (**C**) The graph shows the normalised average recovery profiles of GFP-Cnn (*green*) and Sas-4-mCherry (*red*) 60 s after bleaching (n = 10 centrosomes from 10 embryos). Each profile is normalised so its maximum signal equals one and plotted taking into account the average spatial offset between the two signals—~0.21 μm—see (**H**). The inset image shows the average fluorescent signals of GFP-Cnn (*green*) and Sas-4-mCherry (*red*) overlaid taking into account their average spatial offset. *Figure 1. continued on next page*

*Figure 1. Continued*

(**D–G**) Images (**D**, **F**) and graphs (**E**, **G**) depict the same data as in (**B**) and (**C**) but for either Spd-2-GFP (*green*) and Sas-4-mCherry (*red*) (**D**, **E**) or for Spd-2-GFP (*green*) and RFP-Cnn (*red*) (**F**, **G**). (**H**) The graph shows the position of each GFP signal at 60 s post bleaching relative to the position of the mCherry/RFP signal (always positioned at 0; 0) for each combination of proteins, as indicated. Each dot represents a single centrosome.

The following source data is available for figure 1:

**Source data 1**. Measuring the spatial offset between recovering GFP-Cnn and Sas-4-mCherry, Spd-2-GFP and Sas-4-mCherry, and Spd-2-GFP and RFP-Cnn during S-phase.

microscope. We first examined the centrosomal recruitment of these proteins from the cytosol during S-phase (*Figure 1B,D*; *Videos 1, 2*). Prior to photobleaching, Sas-4-mCherry appeared as a single tight spot, presumably localising to the two centrioles (*Gopalakrishnan et al., 2011*; *Fu and Glover, 2012*; *Mennella et al., 2012*), which cannot be resolved with a standard confocal microscope; GFP-Cnn and Spd-2-GFP occupied a relatively broad area around the centrioles (*Figure 1B,D*; t = −30 s), consistent with their known PCM localisation (*Conduit et al., 2014b*). The centrosomal fluorescence of all three proteins recovered after photobleaching, but we noticed that the recovering GFP and mCherry signals were not aligned in the X-Y plane (*Figure 1B,D*; t = 60–180 s). We plotted the relative positions of the recovering signals after using fluorescent beads to adjust for any systemic shift between the green and red channels (*Figure 1H*; *Figure 1—source data 1*; see 'Materials and methods'), and calculated the average distance between the recovering signals: at 60 s post bleaching, recovering Sas-4-mCherry was offset from recovering GFP-Cnn by an average of ∼0.21 μm (*blue* dots, *Figure 1H*) and from recovering Spd-2-GFP by an average of ∼0.17 μm (*orange* dots, *Figure 1H*). In contrast, the recovering Spd-2-GFP signal was offset from the recovering RFP-Cnn signal by an average of only ∼0.053 μm (*purple* dots, *Figure 1H*). We illustrate these differences by displaying the average fluorescence profiles of each pair of markers offset by the average distance between each marker at 60 s post-bleaching (*Figure 1C,E,G*).

During S-phase, an excess of Sas-4 is recruited to centrioles, and a large fraction of these molecules become irreversibly incorporated into newly forming daughter centrioles, while the remainder is later shed from the centrioles during mitosis (*Novak et al., 2014*). We wondered, therefore, if the offset between the Sas-4-mCherry and GFP-Cnn or Spd-2-GFP recovering signals was a result of

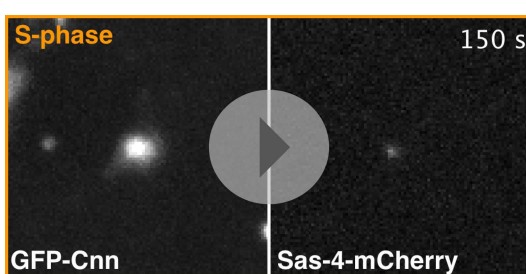

**Video 1.** Recovery dynamics of GFP-Cnn and Sas-4-mCherry during S-phase. This video shows the fluorescent signals of GFP-Cnn (left panel) and Sas-4-mCherry (right panel) recovering during S-phase after photobleaching at t = 0 s. Both signals are detectable 30 s after photobleaching and continue to increase in intensity thereafter. The GFP-Cnn signal initially recovers centrally and then spreads outwards, as described previously (*Conduit et al., 2010*, *2014a*, *2014b*), whereas the Sas-4-mCherry signal recovers as a single tight focus and does not spread outwards.

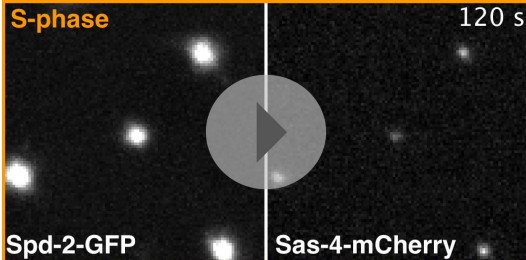

**Video 2.** Recovery dynamics of Spd-2GFP and Sas-4-mCherry during S-phase. This video shows the fluorescent signals of Spd-2-GFP (left panel) and Sas-4-mCherry (right panel) recovering during S-phase after photobleaching at t = 0 s. Both signals are detectable 30 s after photobleaching and continue to increase in intensity thereafter. The Spd-2-GFP signal initially recovers centrally and then spreads outwards, as described previously (*Conduit et al., 2014b*), whereas the Sas-4-mCherry signal recovers as a single tight focus and does not spread outwards.

Sas-4-mCherry being largely recruited to daughter centrioles and GFP-Cnn and Spd-2-GFP being largely recruited around mother centrioles.

As an initial test of this hypothesis, we performed two-colour FRAP experiments in M-phase, when centriole duplication has been completed. After photobleaching, the centrosomal GFP-Cnn and Spd-2-GFP signals recovered immediately, while the Sas-4-mCherry signal only began to recover robustly after the centrosomes separated at the end of mitosis—when a new round of centriole duplication begins (*Figure 2*; *Videos 3, 4*; *Figure 2—source data 1*). These findings are consistent with our hypothesis that Sas-4 molecules are only recruited to growing daughter centrioles, and they strongly suggest that the Cnn and Spd-2 molecules that are recruited around mother centrioles during M-phase are not recruited there as part of a complex with Sas-4.

## Super-resolution microscopy confirms that Sas-4 molecules are recruited exclusively to growing daughter centrioles

In order to directly test if Sas-4 molecules are only recruited to growing daughter centrioles, we turned to 3D-Structured Illumination Microscopy (3D-SIM), which has approximately twice the spatial resolution of standard confocal microscopy. Using 3D-SIM in living embryos, we could clearly distinguish two adjacent Sas-4-GFP foci at individual centrosomes during S-phase (*Figure 3A*, t = −20 s), presumably representing mother-daughter centriole pairs. We combined 3D-SIM with FRAP (*Conduit et al., 2014b*) and found that the Sas-4-GFP signal only recovered at a single foci (*Figure 3A*, t = 120 s to t = 280 s). We confirmed that this recovery occurred at the daughter centriole by performing a two-colour-3D-SIM-FRAP experiment in embryos co-expressing Sas-4-GFP and Asl-mCherry, as Asl forms a toroid around only the mother centrioles (*Figure 3B*, t = −30 s) (*Novak et al., 2014*). Strikingly, the recovering Sas-4-GFP fluorescence always lay outside of the Asl-mCherry toroid (*Figure 3B*, t = 300 s), whereas control unbleached centrosomes still contained two Sas-4-GFP foci, one of which lay inside the Asl-mCherry toroid (*Figure 3C*, t = 300 s). This suggested that new Sas-4 molecules are recruited only to the daughter centrioles. However, during acquisition centrosomes move in the x-y plane, and given that the green and red channels are acquired sequentially on this particular imaging system it is possible for the green and red signals to become misaligned. To be sure the recovering Sas-4-GFP signal represented the daughter centriole, rather than a mis-positioned mother centriole, we therefore measured the distance between the centre of the Asl-mCherry signal and the centres of the pre- and post-bleached Sas-4-GFP signals (*Figure 3D*; *Figure 3—source data 1*). This revealed that the average position of the post-bleached Sas-4-GFP signal closely matched the position of the daughter Sas-4-GFP signal, but not the mother Sas-4-GFP signal (*Figure 3E*), confirming that Sas-4 molecules are recruited only to daughter centrioles. As Asl molecules are known to turn over at the mother centriole at this stage in the cycle (*Novak et al., 2014*), their recruitment cannot occur as part of Sas-4 dependent S-CAP complexes.

We next performed a similar two-colour 3D-SIM FRAP experiment in embryos expressing Spd-2-GFP and Sas-4-mCherry. Here, the recovering Sas-4-mCherry foci lay adjacent to the recovering Spd-2-GFP signal (*Figure 3F*, t = 300 s), which is known to initially recover as a toroid around the mother centriole before spreading outwards in a fibrous network (*Conduit et al., 2014b*). Unbleached centrosomes still contained two Sas-4-mCherry foci, one of which lay at the centre of the Spd-2-GFP network (*Figure 3G*, t = 300 s). Together, these observations demonstrate that in these embryos Sas-4 is only recruited to growing daughter centrioles, while Asl and Spd-2 are recruited only around mother centrioles.

## S-CAP complexes are of low abundance in the early embryonic cytosol

We next analysed the abundance of potential S-CAP complexes in syncytial embryos. We expressed a Sas-4-GFP construct at near endogenous levels in embryos lacking endogenous Sas-4—this construct is functional as it rescues the Sas-4 mutant phenotype (*Novak et al., 2014*). From these embryos we produced extracts where the centrosomes had been removed by centrifugation, immunoprecipitated Sas-4-GFP using anti-GFP coated beads and then examined the relative proportion of bound and unbound S-CAP complex proteins. This approach had two advantages over immunoprecipitating endogenous Sas-4 with anti-Sas-4 antibodies: (1) anti-GFP antibodies are less likely to perturb S-CAP complex assemblies; (2) we could perform negative controls using wild-type extracts that did not contain any Sas-4-GFP protein.

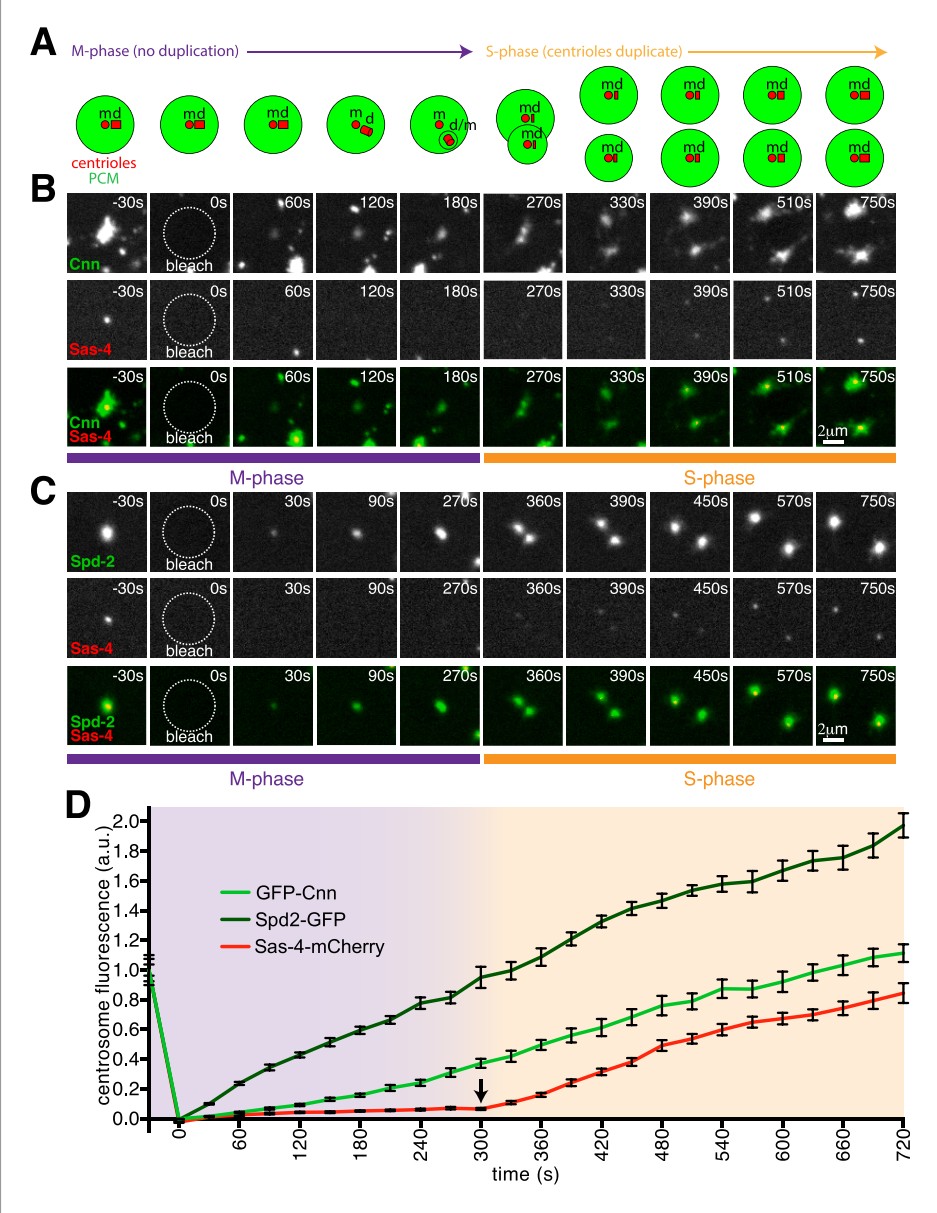

**Figure 2**. Cnn and Spd-2, but not Sas-4, are recruited to centrosomes during M-phase. (**A**) A schematic illustration of the centrosomal events that occur during M-phase (*purple* arrow) and the following S-phase (*orange* arrow) in *Drosophila* syncytial embryos. The mother centriole ('m') organises the PCM (green) and remains 'engaged' to its fully formed daughter centriole ('d') until late M-phase. During late M-phase the centrioles disengage and the daughter centriole matures into a mother and starts to organise its own domain of PCM. At the start of the following S-phase, both the old and new mother centrioles template the formation of a new daughter centriole. (**B**, **C**) Images show how GFP-Cnn (top row in **B**, *green* in bottom row), Spd-2-GFP (top row in **C**, *green* in bottom row) and Sas-4-mCherry (middle rows in **B** and **C**, *red* in bottom rows) fluorescence signals recover after photobleaching during M-phase and the following S-phase. Time in seconds before and after photobleaching (t = 0 s) is shown in the top right of each panel. Note how the GFP-Cnn (**B**) and Spd-2-GFP (**C**) signals recover immediately after photobleaching, but that a recovering Sas-4-mCherry signal can only be detected once the embryos enter the following S-phase (and initiate a new round of centriole duplication). (**D**) A graph showing the fluorescence recovery over time of Sas-4-mCherry (*red*), GFP-Cnn (*light green*) and Spd-2-GFP (*dark green*) relative to their initial fluorescence values. Measurements were taken in the region of the centrosome where the fluorescent signals overlapped. The arrow indicates the sudden change in the recovery dynamics of the Sas-4-mCherry signal.

*Figure 2. continued on next page*

*Figure 2. Continued*

The following source data is available for figure 2:

**Source data 1**. A comparison of the centrosomal fluorescence recovery after photobleaching of GFP-Cnn, Spd-2-GFP and Sas-4-mCherry during M-phase and the following S-phase.

As expected, we saw a strong depletion of Sas-4-GFP from the Sas-4-GFP extract (compare lanes 2 and 4, *Figure 4A*) and a strong enrichment of Sas-4-GFP in the Sas-4-GFP bound sample (lane 6, *Figure 4A*). In contrast, the negative control showed no depletion of endogenous Sas-4 from the wild-type extract (compare lanes 1 and 3, *Figure 4A*), or enrichment of endogenous Sas-4 in the wild-type bound sample (lane 5, *Figure 4A*). Cnn, γ-tubulin, pericentrin-like-protein (PLP) and Asl (the other characterised components of the S-CAP complexes), were not obviously co-depleted with Sas-4-GFP from the extract (compare lanes 2 and 4, *Figure 4B–E*), although we reliably detected small amounts of Cnn and γ-tubulin in the Sas-4-GFP bound samples (compare lanes 5 and 6, *Figure 4B,C*). A small amount of Asl was also detectable in the Sas-4-GFP bound sample, but this was also seen in the control bound sample (compare lanes 5 and 6 in *Figure 4D*). We conclude that only a very small fraction of Cnn and γ-tubulin molecules form cytosolic complexes with Sas-4 in these embryo extracts, while any interaction between Sas-4 and Asl or PLP is undetectable with these methods. Thus, the previously described S-CAP complexes are either absent or present at very low levels in these embryo extracts.

## Concluding remarks

Our results suggest that S-CAP complexes do not play a significant role in mitotic PCM assembly in *Drosophila* syncytial embryos. In support of this conclusion, previous reports have shown that centrioles in *Drosophila* cells lacking cytosolic Sas-4 can recruit PCM during mitosis (*Stevens et al., 2007*; *Riparbelli and Callaini, 2011*), and this also appears to be true in *Caenorhabditis elegans* embryos (*Kirkham et al., 2003*; *Leidel and Gönczy, 2003*) and in HeLa cells (*Kitagawa et al., 2011*). Moreover, SPD-2 and SPD-5, the likely functional homologues of Spd-2 and Cnn, exist mostly as monomers in the cytosol of *C. elegans* embryos and do not detectably interact with Sas-4 (*Wueseke et al., 2014*). Thus, the mechanism of mitotic PCM assembly in flies, worms and human cells does not appear to involve the pre-assembly of Sas-4-dependent cytosolic PCM complexes. Importantly, Sas-4 may have a more indirect role in mitotic PCM assembly in fly embryos as centriolar Sas-4 (as opposed

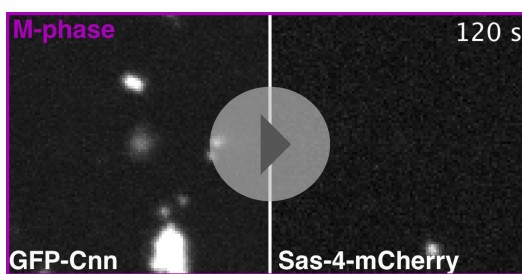

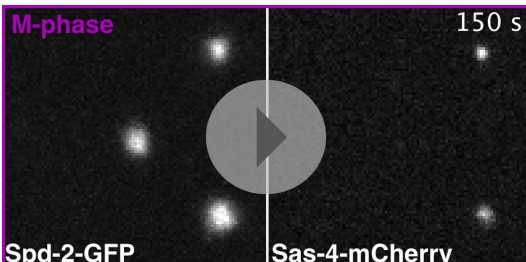

**Video 3.** Recovery dynamics of GFP-Cnn and Sas-4-mCherry during M-phase/S-phase. This video shows the fluorescent signals of GFP-Cnn (left panel) and Sas-4-mCherry (right panel) recovering during M-phase and then during the following S-phase; the centrosome was bleached at t = 0 s in M-phase. The GFP-Cnn signal is detectable 30 s after photobleaching and continues to increase during M-phase and the following S-phase, when the centrosome divides into two. The Sas-4-mCherry signal, however, only becomes detectable 270 s after photobleaching, once the embryo has transitioned from M-phase into the following S-phase.

**Video 4.** Recovery dynamics of Spd-2-GFP and Sas-4-mCherry during M-phase/S-phase. This video shows the fluorescent signals of Spd-2-GFP (left panel) and Sas-4-mCherry (right panel) recovering during M-phase and then during the following S-phase; the centrosome was bleached at t = 0 s in M-phase. The Spd-2-GFP signal is detectable 30 s after photobleaching and continues to increase during M-phase and the following S-phase, when the centrosome divides into two. The Sas-4-mCherry signal, however, only becomes detectable 330 s after photobleaching, once the embryo has transitioned from M-phase into the following S-phase.

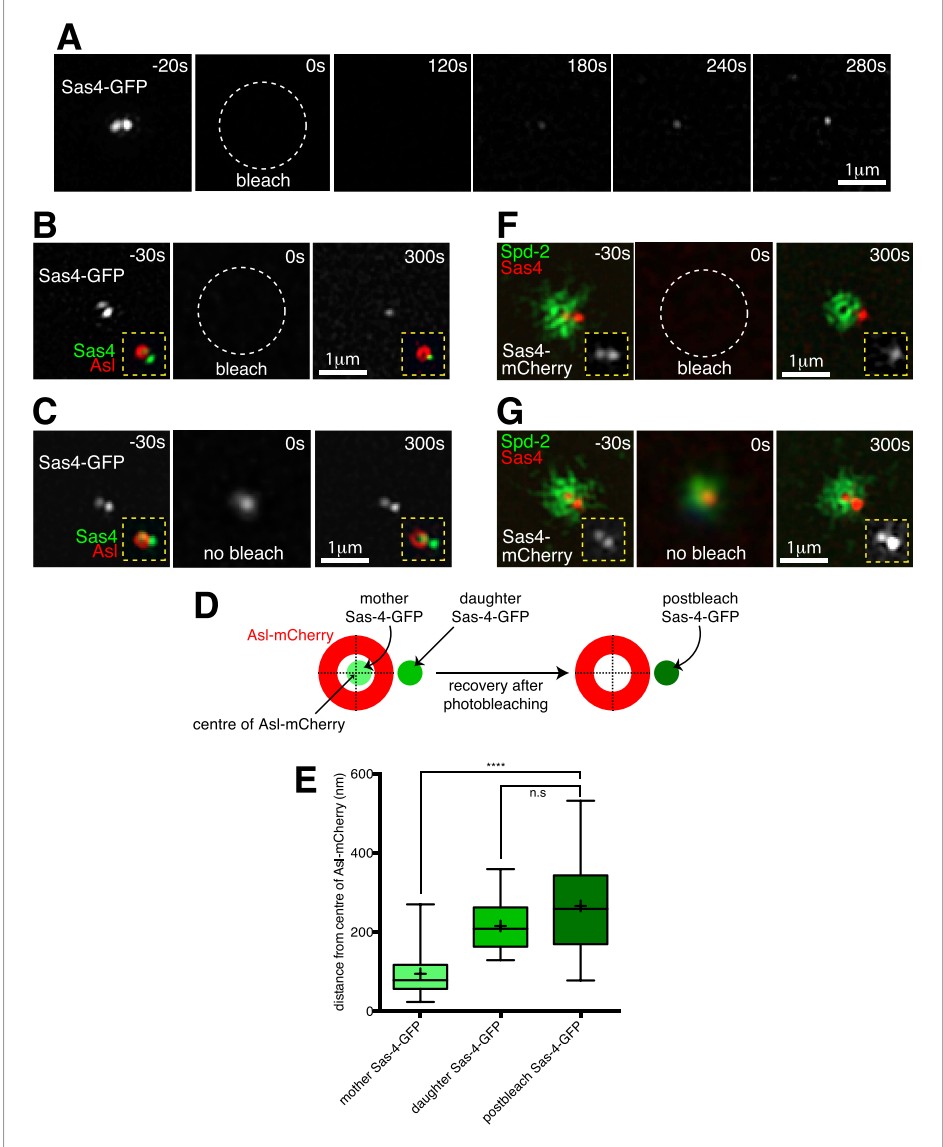

**Figure 3**. Two-colour-3D-SIM FRAP reveals that Sas-4-mCherry is recruited only to growing daughter centrioles, while PCM proteins are recruited only around mother centrioles. (**A**) 3D-SIM images show how during S-phase two adjacent Sas-4-GFP foci can be resolved at an individual centrosome (t = −20 s), and how the Sas-4-GFP fluorescence signal recovers only as a single foci after photobleaching (t = 120 s to t = 280 s). Time in seconds before and after photobleaching (t = 0 s) is shown in the top right of each panel. (**B**) Two-colour 3D-SIM images show how Sas-4-GFP fluorescence recovers relative to Asl-mCherry fluorescence (which surrounds the mother centriole). The Sas-4-GFP is shown in greyscale; insets (*yellow dashed lines*) display the overlay of Sas-4-GFP (*green*) and Asl-mCherry (*red*). Note how after photobleaching the Sas-4-GFP fluorescence recovers outside of the Asl-mCherry toroid. (**C**) Complementary images of a control centrosome adjacent to the one shown in (**B**) where the Sas-4-GFP signal was not photobleached. The t = 0 s panel is a widefield image (see 'Materials and methods'). (**D**, **E**) Schematic (**D**) and box-plot (**E**) show how the average position of the post-bleached Sas-4-GFP signal is similar to the position of the daughter, but not the mother, centriole's prebleached Sas-4-GFP signal, relative to the Asl-mCherry signal. Boxes in **E** extend from the 25th-75th percentiles, whiskers extend from min to max values, lines in boxes are the median values; '+' in boxes are the mean values; n = 25 centrosomes from 4 embryos. **** indicates where p < 0.0001; n.s. indicates where p = 0.09, and is therefore not significant. (**F**) 3D-SIM images show how Sas-4-mCherry (*red*) recovers relative to recovering Spd-2-GFP (*green*). An overlay of Sas-4-mCherry and Spd-2-GFP fluorescence is shown in the main panels; insets (*yellow dashed lines*) display the Sas-4-mCherry signal (greyscale). Note how after photobleaching the Sas-4-mCherry recovers outside of the hollow created by the recovering Spd-2-GFP (t = 300 s). *Figure 3. continued on next page*

*Figure 3. Continued*

(**G**) Complementary images of a control centrosome adjacent to the one in (**F**) where the Sas-4-mCherry and Spd-2-GFP signals were not photobleached. The t = 0 s image is a widefield image, as in (**C**).

The following source data is available for figure 3:

**Source data 1**. Measuring the spatial offset between recovering Sas-4-GFP and Asl-mCherry at super resolution.

to cytoplasmic Sas-4) is required to efficiently recruit Asl molecules around maturing mother centrioles (*Dzhindzhev et al., 2010*; *Novak et al., 2014*), and Asl has an important role in recruiting Spd-2 and Cnn around mother centrioles (*Conduit et al., 2010*, *2014a*).

# Materials and methods

## Transgenic *Drosophila* lines

Sas-4-mCherry and Asl-mCherry P-element-mediated transformation vectors were made by introducing a full length Sas-4 or Asl cDNA into a mCherry C-terminal Gateway vector (*Basto et al., 2008*) downstream of a 2 kb predicted promoter region. Transgenic lines were generated by the Fly Facility in the Department of Genetics, Cambridge, United Kingdom. The other GFP and RFP fusions have been described previously: pUbq-GFP-Cnn (*Lucas and Raff, 2007*), pUbq-Spd-2-GFP (*Dix and Raff, 2007*), Sas-4-GFP (under control of endogenous promoter) (*Novak et al., 2014*) and pUbq-RFP-Cnn (*Conduit et al., 2010*).

## Fly Stocks for live cell microscopy

To examine the dynamics of GFP-Cnn and Sas-4-mCherry at centrosomes, we analysed embryos from mothers expressing GFP-Cnn under the control of the pUbq promoter and Sas-4-mCherry under the control of its endogenous promoter in a cnn$^{f04547}$/cnn$^{HK21}$ and sas-4$^{2214}$/sas-4$^{2214}$ mutant background. To examine Spd-2-GFP and Sas-4-mCherry, we analysed embryos from mothers expressing Spd-2-GFP under the control of the pUbq promoter and Sas-4-mCherry under the control of its endogenous promoter in a sas-4$^{2214}$/sas-4$^{2214}$ mutant background. To examine Spd-2-GFP and RFP-Cnn, we analysed embryos from mothers expressing Spd-2-GFP and RFP-Cnn both under the control of the pUbq promoter in a cnn$^{f04547}$/cnn$^{f04547}$ mutant background. To examine Sas-4-GFP at super-resolution, we analysed embryos from mothers expressing Sas-4-GFP under the control of its endogenous promoter in a sas-4$^{2214}$/sas-4$^{2214}$ mutant background. To examine Sas-4-GFP and Asl-mCherry at super-resolution, we analysed embryos from mothers expressing Sas-4-GFP and Asl-mCherry both under the control of their respective endogenous promoters in a sas-4$^{2214}$/sas-4$^{2214}$ mutant background. To examine Spd-2-GFP and Sas-4-mCherry at super-resolution, we analysed embryos from mothers expressing Spd-2-GFP under the control of the pUbq promoter and Sas-4-mCherry under the control of its endogenous promoter in a sas-4$^{2214}$/sas-4$^{2214}$ mutant background.

## FRAP experiments at standard resolution

Imaging was carried out on a Perkin Elmer Spinning Disk confocal system running Volocity software mounted on a Zeiss Axiovert microscope using a 60×/1.4 NA oil objective. Images shown are maximum intensity projections of 5 z-slices taken 0.5 µm apart. Photobleaching of individual centrosomes was carried out using a combination of a focussed 440 nm laser and a focussed 568 nm laser. ImageJ was used to calculate the distance between the centre points of the recovering green and red signals at 60 s post photobleaching. The images were first scaled up fivefold so that each pixel was divided into 5 × 5 pixels—this allowed a more accurate analysis. The X; Y location of the centre of mass of each signal was calculated by thresholding the image and running the 'analyze particles' (centre of mass) macro on the most central Z plane of the centrosome. To adjust for any residual shift in the green and red channels, we calculated the average distance and direction between the 'green' and 'red' signals coming from subresolution TetraSpeck beads (Life Technologies, United Kingdom) (total of 977 beads analysed from 12 images) and used this to correct the centrosome data for any microscope-induced channel misalignment; the green and red

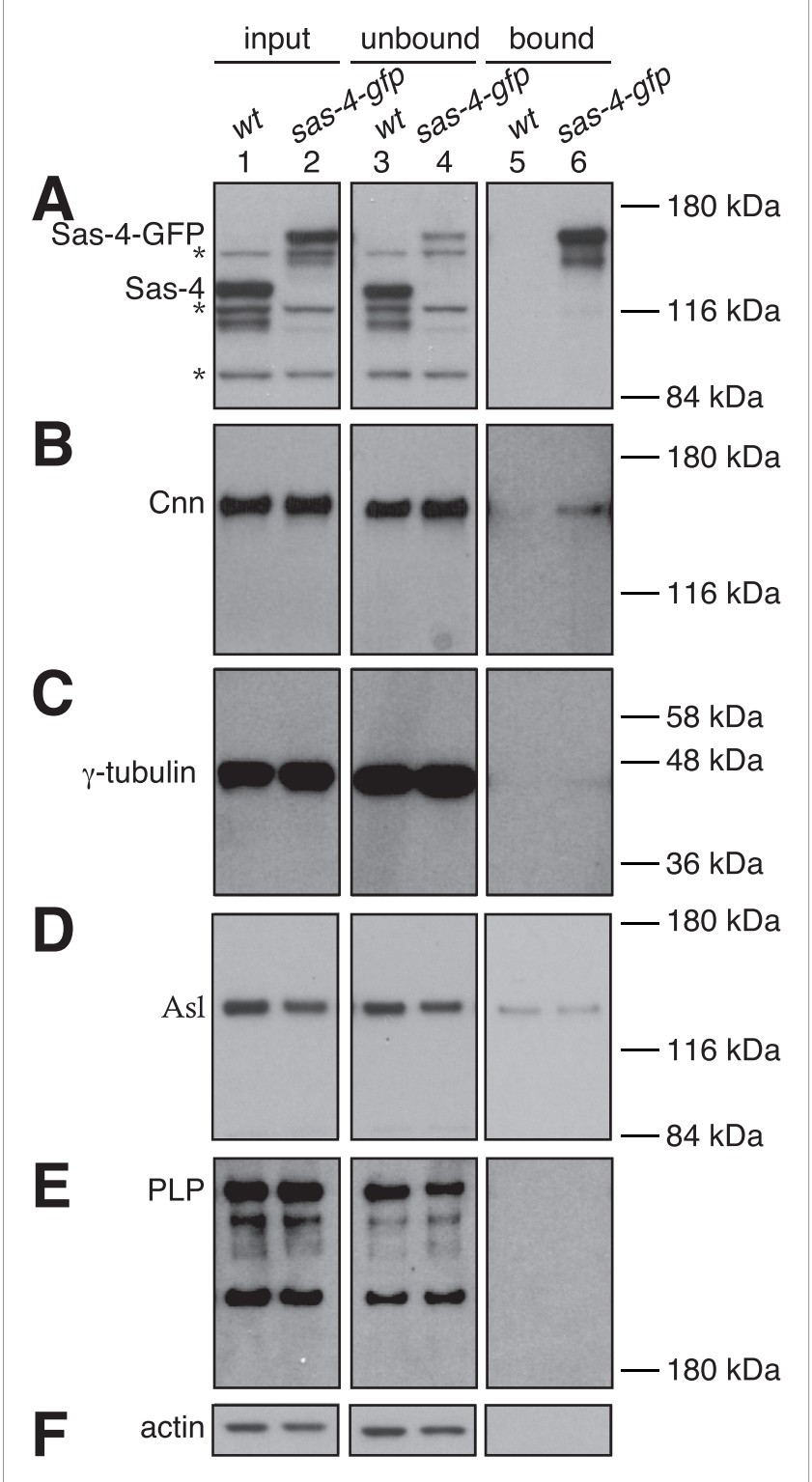

**Figure 4**. Potential S-CAP complexes are of low abundance in Drosophila syncytial embryo extracts. Panels show western blots of anti-GFP immunoprecipitation experiments from WT embryos (lanes 1, 3, 5) or embryos expressing Sas-4-GFP in the absence of endogenous Sas-4 (lanes 2, 3, 6). The membranes were probed for Sas-4 (**A**), Cnn (**B**), γ-tubulin (**C**), Asl (**D**), pericentrin-like-protein (PLP) (**E**) or Actin (as a loading control) (**F**). Lanes 1 and 2 are taken from the initial embryo extracts ('input'); lanes 3 and 4 are 'unbound' samples taken from the extracts after the beads had been removed; lanes 5 and 6 are 'bound' samples taken from the beads after incubation with extract. The * symbols

*Figure 4. Continued*

in (**A**) highlight non-specific bands. Note that the signal intensities can only be directly compared between the 'input' and 'unbound' lanes (see 'Materials and methods').

signals from the beads were offset by a distance of 0.058 µm $\pm$ 0.005 µm. Once corrected, we calculated the distance between the centres of the green and red signals from each centrosome, and then calculated an average distance from all the centrosomes.

ImageJ was used to calculate the 60 s fluorescence recovery profiles of GFP-Cnn, Spd-2-GFP and Sas-4-mCherry from the scaled (5 × 5) images described above. Using the previously calculated centre of mass of each signal, concentric rings (spaced at 0.028 µm and spanning across 3.02 µm) were centred and the average fluorescence around each ring was measured (radial profiling). After subtracting the average cytosolic signal and normalising so the peak intensity of the image was equal to 1, we mirrored the profiles to show a full symmetric centrosomal profile. For each profile, an average distribution from at least 10 centrosomes was calculated. The green and red profiles were plotted on the same graph after manually taking into account the previously calculated average distance between the centre of each signal.

To produce the images that represent the average fluorescent signals at 60 s post bleaching (inset into each graph described above), average projections of green and red images were initially generated separately (after being aligned using the centre of mass coordinates) and then overlaid manually after taking into account the previously calculated average distance between the centre of each signal.

To examine the rate of fluorescence recovery of GFP-Cnn, Spd-2-GFP and Sas-4-mCherry, we measured the green and red fluorescence signals at each timepoint in the pixels where the fluorescent signals overlapped. An ROI was drawn that included all Sas-4-mCherry pixels that had a value above 2 standard deviations from the mean image value, and the total value of the Sas-4-mCherry signal and of the GFP-Cnn or Spd-2-GFP signal within these pixels was calculated. Typically, the ROI was 10 × 10 pixels (1.05 µm × 1.05 µm). The local cytoplasmic background fluorescence was then subtracted from this value. An average value from at least 10 centrosomes was calculated and normalised by dividing it by the average initial pre-bleach value. These normalised average values were then used for each data point in the graph.

## 3D-structured illumination (sub-diffraction resolution) microscopy

Living embryos were imaged at 21°C on a DeltaVision OMX V3 Blaze microscope (GE Healthcare, United Kingdom) equipped with a 60×/1.42 oil UPlanSApo objective (Olympus), 488 nm and 593 nm diode lasers and Edge 5.5 sCMOS cameras (PCO). Spherical aberration was minimized by matching the refractive indices (1.514) of the immersion oil to the sample. 3D-SIM image stacks consisting of 6 z-planes were acquired with 5 phases, 3 angles per image plane and a distance of 0.125 µm between planes. The raw data was computationally reconstructed with SoftWorx 6.1 (GE Healthcare) using Wiener filter settings 0.006 and channel specific optical transfer functions. For two colour 3D-SIM, images from the different colour channels were registered with alignment parameters obtained from calibration measurements with 0.2 µm diameter TetraSpeck beads (Life Technologies) using the OMX Editor software. Images shown are maximum intensity projections of several z-slices. The quality of the reconstructed images was assessed using the SIM-Check ImageJ plugin (*Ball et al., 2015*; http://www.micron.ox.ac.uk/microngroup/software/SIMCheck.php) to ensure proper imaging conditions and to avoid reconstruction artefacts.

To perform 3D-SIM FRAP, we utilized the software development kit from GE Healthcare. This allowed us to create a custom acquisition sequence that first acquired a single Z-stack in 3D-SIM (prebleached image), then performed single or multi spot photobleaching (using the standard OMX galvo scanner TIRF/photo-kinetics module), then performed time lapse imaging in widefield mode (including the photobleached image), and then performed a second 3D-SIM Z-stack (5 min recovery image).

## Immunoprecipitation experiments

0–4 hr embryos were collected from either w[67] (wild-type) mothers or from mothers expressing Sas-4-GFP under the control of its endogenous promoter in a sas-4[2214]/sas-4[2214] mutant background. The Sas-4-GFP construct appears to rescue the sas-4[2214] mutant phenotype when expressed from the endogenous promoter, as the flies are fertile and coordinated (*Novak et al., 2014*). The embryos

were dechorionated using 60% bleach, washed thoroughly with 0.05% Tween-20 in distilled water, flash frozen in liquid nitrogen and stored at −80°C. Centrosome free extracts were prepared by homogenising the frozen embryos in 2 ml per gram IP buffer (50 mM HEPES pH 7.6, 1 mM $MgCl_2$, 1 mM EGTA, 1 mM DTT, 1× Protease inhibitor cocktail [Roche]) and centrifuging twice at 15,000 RCF. The extract was maintained at 4°C during preparation. Protein A Dynabeads (Invitrogen) were covalently coupled to rabbit anti-GFP antibodies (this study) using the $BS^3$ crosslinker (ThermoScientific). The amount of antibody-bead conjugate required to pull out most Sas-4-GFP from the extract was calculated empirically, and equal amounts were used for both Sas-4-GFP and wild-type extracts. Before adding the beads to the lysates, a 10 µl 'input' sample was collected from the extracts and mixed with 10 µl of 2× Laemmli sample buffer. The beads were added and the reaction was incubated overnight at 4°C by rotation. At the end of the incubation, a magnet was used to separate the beads from the extract and a 10 µl 'unbound' sample was collected and mixed with 10 µl of 2× Laemmli sample buffer. The 'input' and 'unbound' samples were of the same volume to ensure that the protein levels in each sample could be directly compared. The beads were washed by re-suspension in PBT three times at room temperature, then washed a further five times with PBT for 10 min by rotation at 4°C. The beads were then boiled for 10 min in 50 µl Laemmli sample buffer to produce a 'bound' sample. 10 µl of each sample was run on a 3–8% Tris-Acetate NuPAGE gel (Life Technologies), western blotted and probed for Sas-4, Cnn, Asl, D-PLP, and γ-tubulin using appropriate antibodies: Primary antibodies: N-terminal rabbit anti-Sas-4 antibodies; N-terminal rabbit anti-Cnn antibodies; C-terminal rabbit anti-Asl antibodies; C-terminal rabbit anti-D-PLP antibodies; mouse anti γ-tubulin antibodies (Sigma); mouse anti-actin (Sigma). Secondary antibodies: HRP-conjugated anti-rabbit or anti-mouse antibodies (Roche). SuperSignal West Femto Maximum Sensitivity Substrate (Thermo Scientific) was used as a chemiluminescent HRP substrate.

## Acknowledgements

PTC was supported by a Sir Henry Dale Fellowship jointly funded by the Wellcome Trust and the Royal Society (105653/Z/14/Z) and by an Issac Newton Trust Research Grant from the University of Cambridge awarded to TTW (RG78799). AW, ZN and JWR were supported by a Senior Investigator Award awarded to JWR and funded by the Wellcome Trust (104575/Z/14/Z). The OMX microscope used in this study is part of the Oxford Micron Advanced Bioimaging Unit supported by a Wellcome Trust Strategic Award (091911).

## Additional information

### Funding

| Funder | Grant reference | Author |
| --- | --- | --- |
| Royal Society | 105653/Z/14/Z | Paul T Conduit |
| Wellcome Trust | 104575/Z/14/Z | Alan Wainman, Zsofia A Novak, Jordan W Raff |
| University of Cambridge | RG78799 | Paul T Conduit, Timothy T Weil |

The funders had no role in study design, data collection and interpretation, or the decision to submit the work for publication.

### Author contributions

PTC, AW, Conception and design, Acquisition of data, Analysis and interpretation of data, Drafting or revising the article; ZAN, Acquisition of data, Analysis and interpretation of data, Drafting or revising the article; TTW, Drafting or revising the article, Contributed unpublished essential data or reagents; JWR, Conception and design, Analysis and interpretation of data, Drafting or revising the article

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
