## [Decision Letter]

Thank you for submitting your work entitled “Re-examining the role of *Drosophila* Sas-4 in centrosome assembly using two-colour-3D-SIM FRAP” for peer review at *eLife*. Your submission has been favorably evaluated by Tony Hunter (Senior Editor) and two reviewers.

The reviewers have discussed the reviews with one another and the Senior Editor has drafted this decision to help you prepare a revised submission.

As you will see, the reviewers felt that the experiments were carefully conducted and in general supported your conclusions, although they had some caveats. Reviewer 1 was more positive than Reviewer 2, but upon discussion they agree that it would be worth publishing an appropriately revised version of your paper in *eLife* as a means of alerting a general audience to the fact that cytoplasmic S-CAP complexes serving as preassembled centrosome building blocks are unlikely to exist in the *Drosophila* embryo. However, a negative result is always difficult to prove unequivocally, and some additional experiments have been suggested that would strengthen your conclusions. Please address the following major points in your revised version.

1) There are concerns raised about the imaging data by Reviewer 1 that need to be addressed.

2) Since you did not exactly repeat the protocols used by Gopalakrishnan et al., further discussion of how the methods differ is needed, and ideally you would run a side-by-side control. The use of sucrose gradients on extracts depleted or not of centrosomes would also be informative. Another possibility would be to use reversible crosslinking in an attempt to stabilize potential S-CAP complexes.

The reviewers have raised a number of other points that you should also consider in your revisions.

Reviewer 1:

In this paper Conduit and colleagues use FRAP experiments in combination with 3D-SIM imaging to study the recruitment dynamics of Sas-4, Cnn, Spd-2 and Als. The authors begin by investigating the centrosomal recruitment of Sas-4, Cnn or Spd-2 in *Drosophila* syncytial embryos. Using two color FRAP experiments, in combination with 3D-SIM FRAP experiments, the authors find that Sas-4 is only recruited to growing daughter centrioles during S-phase and that Cnn and Spd-2 molecules are recruited to mother centrioles in the absence of Sas-4 during mitosis. Finally, the authors use biochemical means to immunopurify S-CAP complexes which have been proposed to contain molecules like Spd-2, Cnn, Asl and Sas-4 and have been proposed to be preassembled protein complexes recruited to centrioles using Sas-4 as a centriole targeting factor. Surprisingly, the authors using well controlled experiments show that preassembled S-CAP complexes are not prominent in these extracts and in combination with the FRAP experiments unlike to be recruited as functional unit to centrioles during the cell cycle.

Overall, this is an interesting paper. It utilizes, FRAP in combination with 3D-SIM to study recruitment properties of various centrosome components, which is technically demanding and sets a new standard for the field (although already used previously by the authors). The work presented here by Conduit and colleagues also challenges the existence of S-CAP complexes, which were proposed, with arguably some surprise, in 2011 in a Nature Communication article by Gopalakrishnan and colleagues to be large cytoplasmic complexes containing several centrosome proteins recruited en masse to the centrosome during its assembly. This study is therefore timely and of broad interest to the field of centrosome biogenesis and function and to a broader extent, based on the methods used to the larger field of cell biology. The authors should address the following minor issues:

1) Some of the 3D-SIM images looks a little strange, in particular the Spd-2 signal. The authors should comment on the measures they have taken to ensure that no 3D-SIM reconstruction artifacts have trickled through. For example, have they run SIM-check on their data to rule out reconstruction artifacts? And ensure proper imaging conditions were used in all cases? Furthermore, the details relating to the 3D-SIM imaging are a little bit scant and would need to be bolstered prior to publication. For example, what was the S/N used, etc.? I do not doubt their results (nor would their conclusions be affected by minor reconstruction artifacts), but I would encourage the authors to add further details on their 3D-SIM imaging and image reconstruction protocols.

2) Concerning the isolation of S-CAP complexes. My understanding is that the protocol used here is different. How does this differ from the previous work from Gopalakrishnan and colleagues? Could it be that other large cytosolic complexes (e.g. S-CAP complexes) are removed during this process? Did the authors ever perform sucrose gradients on extracts depleted or not of centrosomes? It would seem important to clarify what the differences are between the two studies and perhaps include a side-by-side comparison. In the Methods section the authors mention: “however, we cannot rule out the possibility that S-CAP complexes are less stable and possibly fall apart under these conditions, perhaps explaining the difference between our findings and those of [10].” This disclaimer seems out of place...do S-CAP complexes exist or not?

Reviewer 2:

Centrosomes are formed by the recruitment of microtubule-organizing pericentriolar material (PCM) to a pair of centrioles whose duplication once per cell cycle ensures the presence of two microtubule-organizing centers in mitosis. Centriole duplication occurs in a series of steps that have been well characterized at the molecular and latterly also structural level. Sas-4 (also known as CPAP/CENPJ in vertebrates) functions in this pathway at one of the later steps, the formation of the centriolar microtubule wall. Sas-4 has also been implicated in PCM assembly, largely on the basis of its presence in complexes (‘S-CAP complexes’) with key pericentriolar material proteins in the *Drosophila* embryo (10; 11). Here, Conduit and co-workers re-examine the role of Sas-4 in PCM recruitment in the same experimental model. Based on distinct patterns of recruitment observed by high-resolution microscopy, the authors conclude that PCM proteins are not recruited to centrosomes as part of S-CAP complexes and that cytoplasmic Sas-4 is therefore not directly involved in PCM recruitment.

I find it difficult to assess the merit of this manuscript. On the one hand it is well executed and corrects a mistaken notion of Sas-4 as a direct participant in PCM assembly. On the other, this notion never found widespread acceptance in the field because it contradicted what we have known about Sas-4 function for many years from work in *Drosophila* and other experimental models. Further, as discussed below Sas-4 may still interact with PCM components in the context of the centriolar wall, something the manuscript does not address. It is therefore not clear to me that this work makes a significant enough contribution to our understanding of either Sas-4 function or PCM assembly to merit publication in *eLife*.

1) The key observation reported in the current manuscript is that Sas-4 is not coordinately recruited with the PCM components Cnn and Spd2. This indeed runs counter to the idea of cytoplasmic Sas-4-containing ‘S-CAP’ complexes functioning in PCM assembly. However, the best evidence against such a role for Sas-4 is the complete lack of a PCM phenotype in Sas-4-depletions, as reported already in the very first studies on this protein in *C. elegans* ([12]; Leidel et al., 2003) and in numerous other studies since, as also mentioned by the authors in the concluding paragraph.

2) The manner in which the spatial pattern of Sas-4/Cnn/Spd-2 recruitment was analyzed, by FRAP at standard resolution and centroid analysis similar to the way kinetochore protein localization was examined by Ted Salmon and colleagues (Wan et al., 2009) and by 3D-SIM FRAP, is indeed elegant and novel for centrosomes. However, the result comes as no surprise. We know at the very least from the original super-resolution studies in *Drosophila* (16; 9) that PCM proteins including Cnn and Spd-2, but also Asl, localize exclusively around the mother centriole. We also know that Sas-4 on mother centrioles is stably incorporated and does not exchange with the cytoplasmic pool, while daughter centrioles incorporate Sas-4 during S phase (Leidel et al., 2003; [18]). Given those steady state distributions and incorporation patterns, how could the FRAP results be any different?

3) There is little to be gained from the immunoprecipitation data presented in Figure 4, showing weak or no interactions between Sas-4 and Asl, Cnn and γ-tubulin. This is essentially a repeat of previous work by the authors (4), although arguably conducted in a more careful manner. A negative result is always difficult to interpret, even if the authors were to present a positive control (i.e. a protein that does interact with Sas-4 under their conditions). Furthermore, ‘S-CAP’ complexes could represent only a small fraction of the total Cnn or γ-tubulin population, without invalidating the original model of Gopalakrishnan and co-workers.

4) Finally, it should be noted that the most recent work by Gopalakrishnan (24) proposes that the interactions between Sas-4, γ-tubulin and other PCM components occur in the context of the well-described role of Sas-4 as a centriole duplication protein and centriolar microtubule wall component. It is unclear how such interactions could be detected in cytoplasmic extracts, but there is thus a plausible explanation for at least some of the original findings without invoking a direct, coordinated recruitment of Sas-4 with PCM components.

Other points:

1) It should be clarified that the FRAP experiments in this manuscript do not distinguish between recruitment and cytoplasmic turnover.

2) Potentially explained by point 1 above, it is striking that the pattern of Cnn recovery reported here (constant across the cell cycle, Figure 2) does not match the pattern of recruitment reported previously (continuous recruitment in S phase, stable levels in mitosis, Figure 2, [4]).

3) The authors should show that there is no residual shift between red and green channels by imaging the same protein (ideally Sas-4) in both channels.

[Editors' note: further revisions were requested prior to acceptance, as described below.]

Thank you for resubmitting your work entitled “Re-examining the role of *Drosophila* Sas-4 in centrosome assembly using two-colour-3D-SIM FRAP” for further consideration at *eLife*. Your revised article has been favorably evaluated by Tony Hunter (Senior Editor) and two reviewers. The manuscript has been improved but there are some remaining issues that need to be addressed before acceptance, as outlined below:

The reviewers are in general satisfied with your revisions, but indicate that you need to experimentally address the potential for shifts between red and green channels in the SIM/SIM-FRAP experiments. Correcting for chromatic shifts using TetraSpeck beads and then using those same beads to demonstrate an absence of a shift is circular, and does not exclude the possibility of small but significant shifts in the context of the live embryo. Imaging and quantitating the (preferably diffraction limited) signal for the same protein in both channels would eliminate this possibility and increase confidence in the measurements of Sas-4/PCM protein localization/recruitment, which are at the heart of the paper. If you are able to provide such control data, then the paper will be acceptable.

Reviewer 1:

In the revised version of their manuscript the authors have tried to address some of the issues raised in my initial review. They have also addressed some of the issues raised by the other reviewer. Although they have not added much in terms of data they have clarified sufficiently well, I believe, some of the major criticisms. I would still recommend they further discuss the technical differences between their work and that of others, but I would leave this to their discretion and to better contrast the difference observed, and how they relate to models of PCM assembly. I therefore support publication of this manuscript in its current form.

Reviewer 2:

Since my criticisms of the original manuscript by Conduit and Raff centered largely on the significance of their results and thus the suitability of their manuscript for publication in *eLife*, I do not have much to add to my original comments. I agree with Reviewer 1 that there is merit in a paper that clearly refutes the existence of cytoplasmic pre-complexes involving Sas-4 and PCM proteins. However, I fear that S-CAP complexes will persist in the centrosome literature no matter how definitively these complexes are shown not to exist.

With regard to the experimental data and clarifications in the revised manuscript, I'm generally satisfied with those revisions. While a further biochemical characterization of Sas-4 in the cytoplasm along the lines of [23] would be informative, this is clearly beyond the scope of this paper and I agree that sucrose gradients alone would not add significantly to what is primarily a paper built around high-resolution live imaging.

I do, however, think that the authors need to experimentally address the potential for shifts between red and green channels in their SIM/SIM-FRAP experiments. Correcting for chromatic shifts using TetraSpeck beads and then using those same beads to demonstrate an absence of a shift is circular and does not exclude the possibility of small but significant shifts in the context of the live embryo. Imaging and quantitating the (preferably diffraction-limited) signal for the same protein in both channels would eliminate this possibility and increase confidence in their measurements of Sas-4/PCM protein localization/recruitment, which are at the heart of their paper.

---

## [Author Response]

*As you will see, the reviewers felt that the experiments were carefully conducted and in general supported your conclusions, although they had some caveats. Reviewer 1 was more positive than Reviewer 2, but upon discussion they agree that it would be worth publishing an appropriately revised version of your paper in* eLife *as a means of alerting a general audience to the fact that cytoplasmic S-CAP complexes serving as preassembled centrosome building blocks are unlikely to exist in the* Drosophila *embryo. However, a negative result is always difficult to prove unequivocally, and some additional experiments have been suggested that would strengthen your conclusions. Please address the following major points in your revised version.*

1) There are concerns raised about the imaging data by Reviewer 1 that need to be addressed.

Reviewer 1 requested more information about our 3D-SIM imaging methods and we have now supplied these (described in more detail in our response to Reviewer 1).

2) Since you did not exactly repeat the protocols used by Gopalakrishnan et al., further discussion of how the methods differ is needed, and ideally you would run a side-by-side control. The use of sucrose gradients on extracts depleted or not of centrosomes would also be informative. Another possibility would be to use reversible crosslinking in an attempt to stabilize potential S-CAP complexes.

You requested that we further discuss how our biochemical methods differed from those of Gopalakrishnan et al., and that we ideally should run a side-by-side control/comparison using the two methods; you also thought analyzing our extracts on sucrose gradients might be informative. Gopalakrishnan et al. only briefly describe their methods and so we cannot be certain of accurately comparing or reproducing their protocol. Moreover, including such a direct comparison might seem unusually aggressive, so we would prefer not to do this. As you will see, we believe our protocol would very likely detect S-CAP complexes should they be abundant in syncytial fly embryos.

Moreover, the crux of our argument is not whether S-CAP complexes exist or not, but whether S-CAP complexes load Asl, Spd-2 and Cnn into the PCM. Even if Sas-4 does form S-CAP complexes under certain conditions, our microscopy analysis reveals that Sas-4, Asl, Spd-2 and Cnn do not load into the PCM together, so the question over the existence of cytoplasmic S-CAP complexes is not central to the process of PCM assembly.

Regarding the sucrose gradients, we believe that running our extracts on sucrose gradients would not be very informative: Gopalakrishnan et al. detect S-CAP complexes, and they analyze the behavior of these isolated complexes on sucrose gradients; in contrast, we do not detect S-CAP complexes, so cannot analyze their behavior on sucrose gradients.

The reviewers have raised a number of other points that you should also consider in your revisions.

Reviewer 1:

1) Some of the 3D-SIM images looks a little strange, in particular the Spd-2 signal. The authors should comment on the measures they have taken to ensure that no 3D-SIM reconstruction artifacts have trickled through. For example, have they run SIM-check on their data to rule out reconstruction artifacts? And ensure proper imaging conditions were used in all cases? Furthermore, the details relating to the 3D-SIM imaging are a little bit scant and would need to be bolstered prior to publication. For example, what was the S/N used, etc.? I do not doubt their results (nor would their conclusions be affected by minor reconstruction artifacts), but I would encourage the authors to add further details on their 3D-SIM imaging and image reconstruction protocols.

The reviewer thought that our 3D-SIM images of Spd-2-GFP looked strange. We now describe our 3D-SIM acquisition and reconstruction methods in more detail in the Experimental Procedures (subsection “3D-structured illumination (Sub-diffraction resolution) microscopy”). We also provide Figure 5 showing a SIM-Check (a tool developed to specifically analyse the quality of SIM images) analysis of the Spd-2-GFP OMX data. This shows that the modulation contrast-to-noise ratios at centrosomes have scores of ∼12 and ∼11 for pre and postbleach images, respectively—well above the acceptable value of 3, and indicative of high local contrast between the stripes (generated as part of the SI imaging) and the signal. This confirms that the centrosomal Spd-2-GFP signal in the final reconstructed images is likely to represent true localization rather than reconstruction artifacts. We also note that the appearance of Spd-2-GFP in this paper is similar to that in our previous *eLife* paper (Conduit et al., 2014).

Author response image 1.Quality control of Spd-2-GFP 3D-SIM images using SIM-Check.(A, B) Reconstructed images of centrosomes prior to photobleaching (A) and 5 min post photobleaching (B) in an embryo expressing Spd-2-GFP. Left and right insets in each image show a control (unbleached) and the experimental (photobleached) centrosome, respectively. (C, D) Modulation contrast-to-noise ratio (MCNR) images of the raw data used to produce the images and insets in (A) and (B). The Color LUT indicates varying MCNR values, with anything above 3 (purple) being acceptable. The Average feature (centrosomal) MCNR in the prebleached (C) and postbleached (D) images is ∼12 and ∼11, respectively. (E, F) Images of the reconstructed data shown in (A) and (B) colour coded according to the underlying MCNR of the raw data shown in (C) and (D).**DOI:**
http://dx.doi.org/10.7554/eLife.08483.015

*2) Concerning the isolation of S-CAP complexes. My understanding is that the protocol used here is different. How does this differ from the previous work from Gopalakrishnan and colleagues? Could it be that other large cytosolic complexes (e.g. S-CAP complexes) are removed during this process? Did the authors ever perform sucrose gradients on extracts depleted or not of centrosomes? It would seem important to clarify what the differences are between the two studies and perhaps include a side-by-side comparison. In the Methods section the authors mention: “however, we cannot rule out the possibility that S-CAP complexes are less stable and possibly fall apart under these conditions, perhaps explaining the difference between our findings and those of*
[10]*.” This disclaimer seems out of place...do S-CAP complexes exist or not?*

The reviewer asks for a comparison between our IP protocol and that of Gopalakrishnan et al., and asks for clarification as to whether we think S-CAP complexes exist or not. As discussed above, we include a comparison of the methodologies as an Clearly we cannot detect significant levels of these complexes under the conditions we use here, but we cannot directly compare our results to those of Gopalakrishnan et al., who provided no information about the percentage of each cytoplasmic protein that co-immunoprecipitated with Sas-4.

Reviewer 2:

*[…] I find it difficult to assess the merit of this manuscript. On the one hand it is well executed and corrects a mistaken notion of Sas-4 as a direct participant in PCM assembly. On the other, this notion never found widespread acceptance in the field because it contradicted what we have known about Sas-4 function for many years from work in* Drosophila *and other experimental models. Further, as discussed below Sas-4 may still interact with PCM components in the context of the centriolar wall, something the manuscript does not address. It is therefore not clear to me that this work makes a significant enough contribution to our understanding of either Sas-4 function or PCM assembly to merit publication in* eLife*.*

*1) The key observation reported in the current manuscript is that Sas-4 is not coordinately recruited with the PCM components Cnn and Spd2. This indeed runs counter to the idea of cytoplasmic Sas-4-containing ‘S-CAP’ complexes functioning in PCM assembly. However, the best evidence against such a role for Sas-4 is the complete lack of a PCM phenotype in Sas-4-depletions, as reported already in the very first studies on this protein in* C. elegans *(*[12]*; Leidel et al., 2003) and in numerous other studies since, as also mentioned by the authors in the concluding paragraph.*

The reviewer questions the significance of our work, arguing that the idea that S-CAP complexes directly participate in PCM assembly never found widespread acceptance in the field, and pointing out that early studies in worm embryo effectively ruled out the idea that cytoplasmic Sas-4 was required for PCM assembly. We agree that these early worm experiments and our own early experiments in fly embryos (Stevens et al., Curr. Biol. 2007) made an S-CAP model unlikely. Nevertheless, we disagree with the suggestion that the S-CAP model was not taken seriously in the field for the following reasons:

A) Both S-CAP papers were published in high impact journals (Nature Communications and Nature Cell Biology), well after the original papers indicating that this model was unlikely to be correct.

B) We have had trouble publishing several papers in the past few years as reviewers questioned our data because it appeared to contradict the S-CAP model.

C) S-CAP complexes have been discussed as a serious model in virtually every recent review of centrosome/centriole assembly (e.g. Brito et al., Curr. Op. Cell Biol., 2012, Gonczy, Nat. Rev. Mol. Cell Biol., 2012; Mahen et al., Curr. Op. Cell Biol., 2012; Mennella et al., TICB, 2013; Woodruff et al., Philos. Trans. R Soc. Lond. B Biol. Sci. 2014) and also in all four of the first super-resolution papers describing how the PCM is organised around the centrioles (Mennella et al., NCB; Lawo et al., Curr. Biol.; Fu et al., Open Biol.; Sonnen et al., Biol. Open - all 2012).

D) In the paper that fails to detect cytoplasmic S-CAP complexes in worm embryos (Wueseke et al., Mol. Biol. Cell, 2014), the authors conclude that more work is required to assess whether this is due to species-specific differences between worms and flies.

*2) The manner in which the spatial pattern of Sas-4/Cnn/Spd-2 recruitment was analyzed, by FRAP at standard resolution and centroid analysis similar to the way kinetochore protein localization was examined by Ted Salmon and colleagues (Wan et al., 2009) and by 3D-SIM FRAP, is indeed elegant and novel for centrosomes. However, the result comes as no surprise. We know at the very least from the original super-resolution studies in* Drosophila *(*[16]*;*
[9]*) that PCM proteins including Cnn and Spd-2, but also Asl, localize exclusively around the mother centriole. We also know that Sas-4 on mother centrioles is stably incorporated and does not exchange with the cytoplasmic pool, while daughter centrioles incorporate Sas-4 during S phase (Leidel et al., 2003;*
[18]*). Given those steady state distributions and incorporation patterns, how could the FRAP results be any different?*

The reviewer points out that several studies had previously concluded that Sas-4 is irreversibly incorporated into daughter centrioles during their formation, while other studies had concluded that Asl, Cnn and Spd-2 localise around mother centrioles. Thus, one could already infer that Sas-4 does not shuttle these proteins to the mother centriole. The Reviewers’ description of Sas-4 dynamics in fly embryos is, however, not quite correct. In the paper to which the reviewer refers (18) we showed (using a microscope with standard resolution) that an excess of Sas-4 builds up at centrioles during S-phase (when centrosomes are growing in size) and this excess is then gradually lost during mitosis (when centrosomes have reached their full size). Thus, although a significant fraction of Sas-4 molecules are clearly incorporated irreversibly into daughter centrioles, it was entirely possible that the excess Sas-4 molecules (that are not irreversibly incorporated into daughters) could have been shuttling S-CAP complexes to mother centrioles. Our super-resolution studies now prove that this is not the case. We clarify this point in the revised manuscript (subsection “The centrosomal recruitment of Sas-4, Cnn and Spd-2 differs in space and time”).

*3) There is little to be gained from the immunoprecipitation data presented in*
Figure 4*, showing weak or no interactions between Sas-4 and Asl, Cnn and γ-tubulin. This is essentially a repeat of previous work by the authors (*[4]*), although arguably conducted in a more careful manner. A negative result is always difficult to interpret, even if the authors were to present a positive control (i.e. a protein that does interact with Sas-4 under their conditions). Furthermore, ‘S-CAP’ complexes could represent only a small fraction of the total Cnn or γ-tubulin population, without invalidating the original model of Gopalakrishnan and co-workers.*

As we discuss above, we agree with the reviewer that our biochemical analysis adds little definitive information, as these studies can neither prove nor disprove the S-CAP model. Nevertheless, we thought it important to include these studies to show that we do not detect S-CAP complexes under the conditions we normally use when looking for protein-protein interactions. The results, although negative, support our imaging data and we suspect that not including the data would lead readers to wonder what proportion of Asl, Cnn and Spd-2 molecules could be part of S-CAP complexes (something that Gopalakrishnan et al. did not address).

*4) Finally, it should be noted that the most recent work by Gopalakrishnan (*[24]*) proposes that the interactions between Sas-4, γ-tubulin and other PCM components occur in the context of the well-described role of Sas-4 as a centriole duplication protein and centriolar microtubule wall component. It is unclear how such interactions could be detected in cytoplasmic extracts, but there is thus a plausible explanation for at least some of the original findings without invoking a direct, coordinated recruitment of Sas-4 with PCM components.*

The reviewer suggests that the recent paper from the Gopalakrishnan lab (24) shows how Sas-4 could play a role in “tethering” PCM components to the centriole wall. We entirely agree that Sas-4 in the centriole is likely to play an important part in tethering the PCM, and our own experiments strongly support this idea, as we find that the Sas-4 already at the centriole helps recruit Asl to centrioles (18), which then helps recruit the PCM (Conduit et al., 2014). However, this is a completely different concept from the S-CAP model proposed by Gopalakrishnan and colleagues – a model which is not disputed by Zheng et al.

Other points:

1) It should be clarified that the FRAP experiments in this manuscript do not distinguish between recruitment and cytoplasmic turnover.

The reviewer points out that we need to clarify that our FRAP experiments do not distinguish between recruitment and turnover. Given how the Reviewer uses the term “recruitment” in point 2 below, we suspect that he/she uses “recruitment” to mean the net accumulation of molecules at the centrosome and “turnover” to mean the exchange of centrosome molecules for cytoplasmic molecules. We agree that our FRAP data does not distinguish between these two phenomena. However, we use the term recruitment to include both of these phenomena (i.e. we count a molecule as being “recruited” to centrosomes every time a molecule is added to centrosomes from the cytoplasm, irrespective of whether this adds a new molecule to the centrosome or if it replaces an existing molecule). We now explicitly define how we use the term recruitment (Results, first paragraph).

*2) Potentially explained by point 1 above, it is striking that the pattern of Cnn recovery reported here (constant across the cell cycle,*
Figure 2*) does not match the pattern of recruitment reported previously (continuous recruitment in S phase, stable levels in mitosis,*
Figure 2*,*
[4]*).*

The reviewer believes that the pattern of GFP-Cnn recruitment in the current study does not match that reported in our 2010 paper. This is not the case, and we believe this confusion arises from our different definitions of recruitment (see above). In our earlier paper we showed that the total number of GFP-Cnn molecules at centrosomes increases during S-phase (for the reviewer this is “recruitment”) and then stays constant during M-phase (so the reviewer concludes that no recruitment occurs during M-phase). We now show that GFP-Cnn fluorescence recovers after bleaching during both S- and M-phase. Using the reviewers’ definitions, this is due to both recruitment and turnover in S-phase, but only due to turnover in M-phase. However, by our definition, GFP-Cnn molecules are still being recruited to centrosomes during M-phase. Thus, the pattern of GFP-Cnn dynamics in the current study is fully compatible with our earlier analysis. We hope that defining our use of the term “recruitment” helps to clarify this point (Results, first paragraph).

3) The authors should show that there is no residual shift between red and green channels by imaging the same protein (ideally Sas-4) in both channels.

The reviewer thought that we should show that there is no shift between the red and green channels on our confocal microscope by imaging the same protein (ideally Sas-4) in both channels. We apologise for not making this clear but we have performed this control using TetraSpeck beads that fluoresce in both red and green channels; this is the “gold standard” for channel alignment. We have now clarified this point in both the main text (Results, second paragraph) and the Experimental Procedures (subsection “FRAP experiments at standard resolution”).

[Editors' note: further revisions were requested prior to acceptance, as described below.]

Reviewer 1:

In the revised version of their manuscript the authors have tried to address some of the issues raised in my initial review. They have also addressed some of the issues raised by the other reviewer. Although they have not added much in terms of data they have clarified sufficiently well, I believe, some of the major criticisms. I would still recommend they further discuss the technical differences between their work and that of others, but I would leave this to their discretion and to better contrast the difference observed, and how they relate to models of PCM assembly. I therefore support publication of this manuscript in its current form.

Although Reviewer 1 thought that we should further discuss the differences between our work and that of others, he/she kindly left this to our discretion and supported publication of the manuscript in its current form. We agree that an extensive discussion would be of interest to the community, but it is always difficult to accurately assess why two different groups have performed similar experiments but reached different conclusions. We highlight that there are important differences between the two studies, but we prefer to let readers reach their own opinion as to why this might be the case. We have therefore left the paper unchanged in this regard.

Reviewer 2:

Since my criticisms of the original manuscript by Conduit and Raff centered largely on the significance of their results and thus the suitability of their manuscript for publication in eLife, I do not have much to add to my original comments. I agree with Reviewer 1 that there is merit in a paper that clearly refutes the existence of cytoplasmic pre-complexes involving Sas-4 and PCM proteins. However, I fear that S-CAP complexes will persist in the centrosome literature no matter how definitively these complexes are shown not to exist.

*With regard to the experimental data and clarifications in the revised manuscript, I'm generally satisfied with those revisions. While a further biochemical characterization of Sas-4 in the cytoplasm along the lines of*
[23]
*would be informative, this is clearly beyond the scope of this paper and I agree that sucrose gradients alone would not add significantly to what is primarily a paper built around high-resolution live imaging.*

I do, however, think that the authors need to experimentally address the potential for shifts between red and green channels in their SIM/SIM-FRAP experiments. Correcting for chromatic shifts using TetraSpeck beads and then using those same beads to demonstate an absence of a shift is circular and does not exclude the possibility of small but significant shifts in the context of the live embryo. Imaging and quantitating the (preferably diffraction limited) signal for the same protein in both channels would eliminate this possibility and increase confidence in their measurements of Sas-4/PCM protein localization/recruitment, which are at the heart of their paper.

Reviewer 2 thought that we needed to experimentally address the potential for shifts between red and green channels in our SIM/SIM-FRAP experiments, pointing out that the use of TetraSpeck beads does not exclude the possibility of small but significant shifts in the context of the living embryo. To eliminate this possibility, the reviewer suggested that we image the signal from the same protein in both channels to confirm that they align. From these comments we realize that we originally misunderstood the reviewers concern: we thought we needed to address whether the red and green channels were systemically misaligned (which is addressed by the TetraSpeck bead experiment), but the reviewer was actually asking us to address whether the two channels were misaligned because the centrioles have moved slightly during acquisition. The reviewer is correct that this is an important issue, and we apologise that we originally misunderstood this point.

The centrioles can indeed move slightly between the acquisition of the green and red channels (which, for technical reasons, have to be acquired sequentially). This slight movement does not, however, affect the interpretation of our data. To prove this point, we now measure the distance between the center of the Asl toroid (marking the center of the mother centriole acquired in the red channel) and the center of each Sas-4 dot (marking the center of both the mother and daughter centrioles, acquired in the green channel). Prior to photobleaching, one Sas-4 centriole (the presumptive mother) is on average ∼80nm from the center of the Asl toroid, while the other (the presumptive daughter) is on average ∼210nm from the center of the Asl toroid. Thus, the mother centriole shifts by an average of ∼80nm during the time between acquisition of the two channels. After photobleaching, the Sas-4 signal recovers at an average distance of ∼260nm from the center of the Asl toroid: statistically, this distance is significantly different from the average distance the mother centriole shifts during acquisition (∼80nm) but is not significantly different from the average distance between the Asl toroid and the daughter centriole (∼210nm). This new analysis (shown in new Figure 3) strongly supports our conclusion that Sas-4 is recruited to daughter centrioles rather than mother centrioles.